# Surf3R: Rapid Surface Reconstruction from Sparse RGB Views in Seconds

## Abstract

Current multi-view 3D reconstruction methods rely on accurate camera calibration and pose estimation, requiring complex and time-intensive pre-processing that hinders their practical deployment. To address this challenge, we introduce **Surf3R**, an end-to-end feedforward approach that reconstructs 3D surfaces from sparse views without estimating camera poses and completes an entire scene in **under 10 seconds**. Our method employs a multi-branch and multi-view decoding architecture in which multiple reference views jointly guide the reconstruction process. Through the proposed branch-wise processing, cross-view attention, and inter-branch fusion, the model effectively captures complementary geometric cues without requiring camera calibration. Moreover, we introduce a D-Normal regularizer based on an explicit 3D Gaussian representation for surface reconstruction. It couples surface normals with other geometric parameters to jointly optimize the 3D geometry, significantly improving 3D consistency and surface detail accuracy. Experimental results demonstrate that **Surf3R** achieves state-of-the-art performance on multiple surface reconstruction metrics on ScanNet++ and Replica datasets, exhibiting excellent generalization and efficiency.

## 1 Introduction

3D surface reconstruction is a long-standing problem that aims to create 3D surfaces of an object or scene captured from multiple viewpoints Broadhurst et al. (2001); Kutulakos & Seitz (1999); Seitz & Dyer (1999). This technique has wide applications in robotics, graphics, virtual reality, and other fields. Traditional 3D surface reconstruction methods typically include two main approaches: Structure-from-Motion (SfM) combined with Multi-View Stereo (MVS) Chen et al. (2023); Huang et al. (2024); Chen et al. (2024a) or volumetric methods Zak Murez & Rabinovich (2020); Sun et al. (2021). The SfM+MVS pipeline involves first estimating camera poses and generating sparse 3D point clouds using SfM Crandall et al. (2013); Charatan et al. (2024a); Schönberger & Frahm (2016). This is followed by computing per-view depth maps and fusing them into the final 3D surface using MVS techniques Huang et al. (2024); Guédon & Lepetit (2024); Chen et al. (2024a). On the other hand, volumetric methods, such as Atlas Zak Murez & Rabinovich (2020) and NeuralRecon Sun et al. (2021), predict 3D volumes like Truncated Signed Distance Function (TSDF) from multiple views, often avoiding the explicit depth map computation. Although these two kinds of methods achieve high-quality surface reconstruction, they often rely on prior knowledge or require nontrivial pre-processing steps, such as SfM to estimate camera intrinsics and extrinsics. These pre-processing steps often require heavy GPU computation and are time-consuming (typically taking 1–2 hours per scene on a modern GPU), making real-time inference challenging and reducing their practical usability.

To address the aforementioned limitations, inspired by DUSt3R Wang et al. (2024), we propose **Surf3R**, the first feed-forward network that performs pose-free surface reconstruction from sparse RGB inputs in a single pass. Specifically, we first encode all input views using a shared encoder to extract multi-scale visual features. To effectively model cross-view information interactions, we introduce Feature-Refine (FR) blocks that jointly learn not only the pairwise relationships between a selected reference view and all other source views, but also the interactions among source views themselves. When reconstructing a large scene from sparse multi-view images, the geometric correspondence between a selected reference view and certain source views could be insufficient. This is because substantial changes in camera poses make it difficult to directly infer the relation

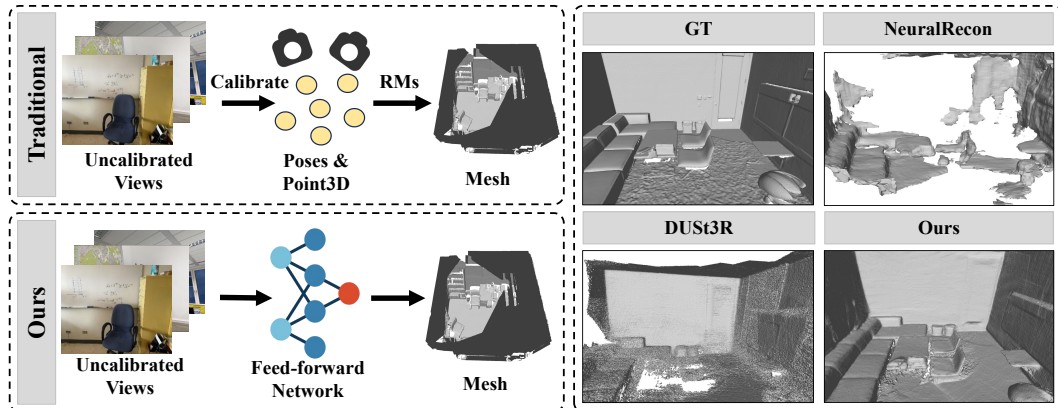

Figure 1: Comparison Between Traditional Methods and Our Approach. Traditional methods rely on SfM for sparse point clouds and calibrated poses, followed by different 3D Reconstruction Methods (RMs). In contrast, our method directly reconstructs the scene from uncalibrated images in **under 10 s**, eliminating the need for calibration or iterative refinement.

between the reference view and those source views. To mitigate this issue, we further introduce a cross-reference fusion mechanism, implemented via a multi-branch design where multiple reference views are independently selected. Each branch processes the input views through its own FR blocks and integrates information using dedicated Cross-Reference Fusion (CRF) blocks, enabling effective propagation of long-range and complementary information across views. Based on the fused multi-view features, we first generate a sparse 3D point cloud for reconstruction. While directly converting this point cloud into a mesh using NKSR Huang et al. (2023) is feasible, our experiments (see Sec. 4.2) show that this naive approach yields poor reconstruction quality. The underlying limitation is that point-cloud supervision is applied in a view-separated manner and thus lacks global 3D consistency. We therefore adopt a Gaussian representation: each Gaussian resides in a unified 3D space and is projected into every view during rendering, so the per-view loss implicitly regularizes the entire scene and yields smoother, more accurate surfaces. The final per-pixel Gaussian primitives are derived from specifically designed Gaussian heads.

To further facilitate accurate surface reconstruction from the predicted Gaussian parameters, we introduce a Depth-Normal Regularization strategy Chen et al. (2024b) designed to enhance the geometric fidelity of the reconstructed surfaces. Specifically, we first apply a flattening operation to the Gaussian primitives to better align them with the local surface geometry. Subsequently, we introduce a D-Normal formulation, in which surface normals are not directly blended from 3D Gaussians, but instead derived from the gradient of the rendered depth map. As a result, it allows the Gaussian parameters to be directly supervised by surface normals, jointly optimizing the geometry and markedly enhancing 3D consistency and surface detail. Extensive experiments on the ScanNet++ datasets demonstrate that Surf3R achieves state-of-the-art surface reconstruction performance. It significantly outperforms both optimization-based and feed-forward baselines in terms of accuracy and completeness. Moreover, when evaluated on the unseen Replica dataset in a zero-shot setting, our model maintains competitive accuracy, demonstrating robust generalization to novel scenes. And it also delivers strong performance on novel view synthesis tasks. Notably, **Surf3R reconstructs an entire scene in under 10 seconds**, making it highly efficient for real-time or interactive applications. We summarize our contributions as follows:

- We present **Surf3R**, the first feed-forward network for pose-free surface reconstruction from sparse multi-view RGB inputs. It achieves real-time surface reconstruction in **under 10 seconds**, offering both high efficiency and scalability.

- We employ a multi-branch architecture where multiple reference views are jointly leveraged to capture long-range cross-view interactions. Furthermore, we introduce a Depth-Normal Regularization strategy to enhance geometric fidelity.

- Extensive experiments on ScanNet++ and Replica datasets demonstrate that Surf3R achieves state-of-the-art surface reconstruction, generalizes zero-shot to new scenes, and remains competitive for novel-view synthesis.

## 2 RELATED WORKS

**Multi-View Surface Reconstruction.** Multi-view surface reconstruction recovers dense geometry from images captured at multiple viewpoints. Classical pipelines fuse depth maps obtained by multi-view stereo Furukawa et al. (2015); Seitz et al. (2006); Schönberger et al. (2016); Zhang et al. (2020); Yao et al. (2018); Bleyer et al. (2011) or optimize voxel occupancy fields Bonet & Viola (1999); Kutulakos & Seitz (2000); Broadhurst et al. (2001); Seitz & Dyer (1999), but these approaches are limited by memory and cross-view noise Tulsiani et al. (2017); Ummenhofer et al. (2017). Implicit neural representations such as signed distance fields Park et al. (2019); Liu et al. (2020); Ma et al. (2023); Sitzmann et al. (2020) and neural volume rendering Yariv et al. (2021); Wang et al. (2021b) alleviate some constraints, yet methods like NeuS Wang et al. (2021b), MonoSDF Yu et al. (2022), and Geo-NeuS Fu et al. (2022) remain optimization intensive and do not scale well. 3D Gaussian Splatting (3DGS) Kerbl et al. (2023) introduces an explicit alternative that rasterizes anisotropic Gaussians in real time Yifan et al. (2019), though the vanilla version lacks geometric supervision. Extensions such as VastGaussian Lin et al. (2024), SuGaR Guédon & Lepetit (2024) and HRGS Li et al. (2025) incorporate view-consistent depth and normal constraints, substantially improving accuracy and convergence. However, these methods often require pre-processing step, limiting their practical deployment. To bridge this gap, we present a feedforward network that dispenses with costly pre-processing while reconstructs high idelity 3D susrface from sparse views in under 10 seconds.

**Novel View Synthesis.** Novel-view synthesis has evolved from geometry-aware volumetric grids, such as Soft3D Penner & Zhang (2017) and voxel colouring Seitz & Dyer (1999), to neural radiance fields like NeRF Mildenhall et al. (2021) and Mip-NeRF Barron et al. (2021), which offer high fidelity in rendering. However, these methods are limited by the computational cost of dense ray sampling, which hampers their scalability and real-time performance. To address this issue, recent advancements in hash-encoded feature grids and sparse voxel accelerations, such as Instant-NGP Müller et al. (2022) and Plenoxels Fridovich-Keil et al. (2022), and KiloNeRF Reiser et al. (2021), have been proposed. These approaches improve runtime efficiency, though they still face challenges in sparse-view or large-scale settings Garbin et al. (2021); Chen et al. (2022). In contrast, 3D Gaussian Splatting (3DGS) Kerbl et al. (2023) achieves real-time rendering by rasterizing anisotropic Gaussians, providing both faster and higher-quality results compared to neural radiance fields. In this work, we exploit the benefits of Gaussian Splatting for surface reconstruction and incorporate normal priors to guide the process, further enhancing the geometric accuracy and detail of the reconstructed surfaces.

**Learning-based 3D Reconstruction.** Learning-based 3D reconstruction approaches have recently seen a lot of progress. Notably, DUSt3R Wang et al. (2024) generates two point maps for an input image pair within a shared coordinate system, implicitly incorporating both intrinsic and extrinsic camera parameters. Nevertheless, the method is intrinsically pair-wise and lacks native support for multi-view input. Multi-view operation is possible only by appending a separate global-alignment stage that registers the individual reconstructions. This alignment must be performed offline, incurs substantial computational overhead, and cannot adapt on-the-fly when additional views become available. Several recent studies Leroy et al. (2024); Wang & Agapito (2024); Zhang et al. (2025); Wang et al. (2025) replace DUSt3R's test-time optimization with feed-forward neural networks to accelerate inference, yet their primary focus lies in view synthesis and related tasks rather than high-fidelity surface reconstruction. In this work, we close this gap by introducing a feed-forward, geometry-aware framework that delivers state-of-the-art surface reconstruction while retaining real-time efficiency.

## 3 METHODOLOGY

Our proposed **Surf3R** framework enables accurate surface reconstruction from a sparse set of RGB images, without the need for known camera intrinsics or poses. First, we introduce a feedforward network that extracts multi-view features and predicts per-pixel 3D Gaussian parameters (Sec.3.1). These Gaussians are then flattened into 2D planes to better represent the surface, which helps achieve more accurate depth estimation for reconstruction (Sec.3.2). Finally, the model is optimized using geometry-aware loss functions (Sec. 3.3).

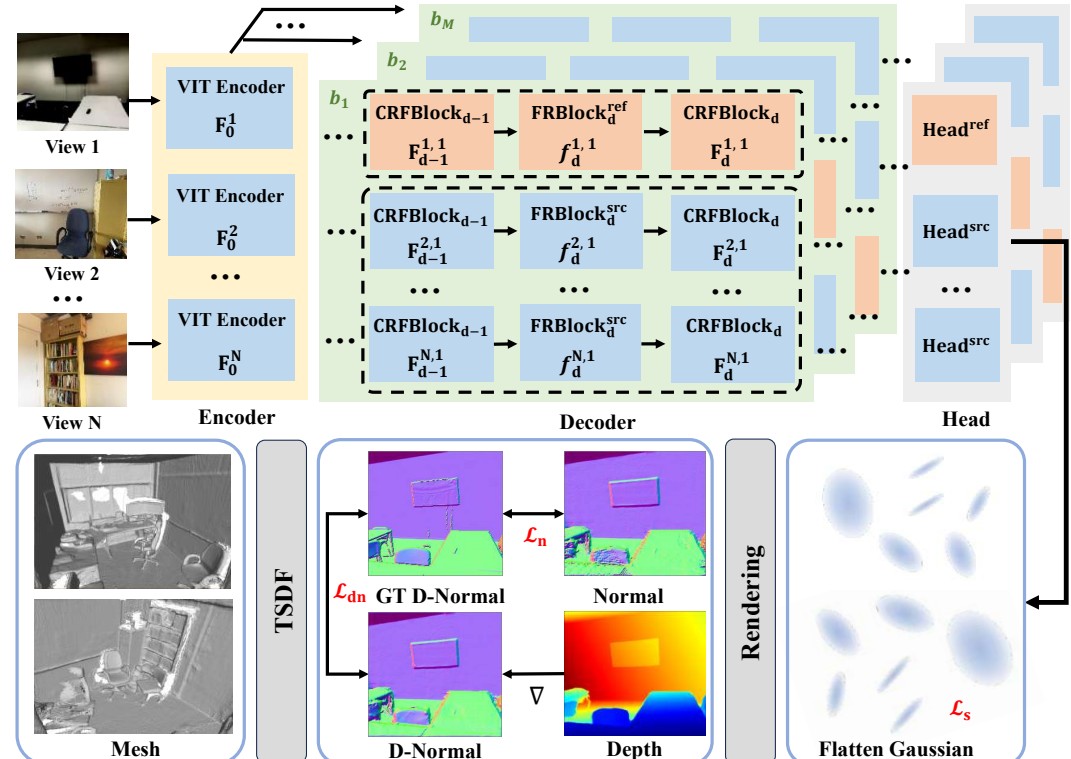

Figure 2: Overall Framework of **Surf3R**. The reference view branches are shown in orange, while the branches of other source views are shown in blue. Each model path uses a different reference view. For clarity, only one of the stacked **FRBlock** and **CRFBlock** is displayed.

## 3.1 FEEDFORWARD GEOMETRY RECONSTRUCTION

As shown in Fig. 2, Surf3R employs a feedforward network architecture to extract and fuse visual information across multiple input views without camera intrinsics and poses. Unlike traditional methods, which depend on estimating these parameters, Surf3R employs a single feedforward pass to directly reconstruct 3D surfaces through view-specific processing.

**Multi-view Feature Encoding and Fusion.** Given $N$ input images $\{I_i\}_{i=1}^N$, a shared-weight Vision Transformer (ViT) is first employed to encode each image $I_i$ into visual tokens $F_0^i = \text{ViT}(I_i)$, with a spatial resolution reduced by a factor of 16. To mitigate the limitation that a single reference view may not provide uniformly accurate geometric cues across the entire scene, we introduce a multi-branch architecture in which multiple reference views collaboratively participate in the reconstruction process. Specifically, we select $M$ reference views $\{r_m\}_{m=1}^M$ to construct $M$ decoding branches $\{b_m\}_{m=1}^M$, each centered around a different reference view. Within each branch $b_m$, a dedicated multi-view decoder, consisting of $D$ stacked Feature-Refine (FR) blocks, is employed to refine features via cross-view attention. These blocks are denoted as $\text{FRBlock}_d^{\text{ref}}$ for the reference view $r_m$ and $\text{FRBlock}_d^{\text{src}}$ for the remaining $N-1$ source views, where $d \in \{1, \ldots, D\}$. At each decoding layer $d$, the token representations are updated in a view-specific manner. For a given view $I_v$, the decoder block takes as input the primary tokens $F_{d-1}^{v,m}$ from view $I_v$ and the secondary tokens $\mathcal{F}_{d-1}^{-v,m} = \{F_{d-1}^{i,m} \mid i \neq v\}$ from all other views. The update is performed as:

$$f_d^{v,m} = \begin{cases} \text{FRBlock}_d^{\text{ref}}(F_{d-1}^{v,m}, \mathcal{F}_{d-1}^{-v,m}), & \text{if } v = r_m \\ \text{FRBlock}_d^{\text{src}}(F_{d-1}^{v,m}, \mathcal{F}_{d-1}^{-v,m}), & \text{otherwise.} \end{cases} \quad (1)$$

To further enhance the expressiveness of feature representations, we introduce a Cross-Reference Fusion (CRF) block after each decoder block to fuse and update per-view tokens computed under different reference views. Specifically, the updated feature is computed as:

$$\mathcal{F}_d^{v,m} = \text{CRFBlock}_d(f_d^{v,m}, f_d^{v,-m}), \quad (2)$$

where $f_d^{v,-m} = \{f_d^{v,1}, \ldots, f_d^{v,m-1}, f_d^{v,m+1}, \ldots, f_d^{v,M}\}$ denotes the set of representations for view $v$ at layer $d$ across all other reference branches.

**Gaussian Parameterizing.** Based on the fused multi-view features, we derive a sparse 3D point cloud. Direct meshing with NKSR Huang et al. (2023) is possible, but Sec. 4.2 shows that it yields poor surface fidelity because global 3D consistency is absent. Accordingly, we adopt a unified 3D Gaussian representation whose projections into all views let per-view losses regularize the entire scene, producing smoother and more accurate surfaces. To predict the final Gaussian parameters from $F_D^{v,m}$, we introduce two types of heads: Head$^{\text{ref}}$ for the reference view and Head$^{\text{src}}$ for the remaining source views. Each head comprises two sets of regression branches. The first set includes a pointmap head and a confidence head, which respectively predict a 3D pointmap $P^{v,m} \in \mathbb{R}^{H \times W \times 3}$ and a confidence map $C^{v,m} \in \mathbb{R}^{H \times W}$ for each view. The second set consists of Gaussian-specific heads that regress the per-pixel Gaussian parameters, including scaling factors $S^{v,m} \in \mathbb{R}^{H \times W \times 3}$, rotation quaternions $q^{v,m} \in \mathbb{R}^{H \times W \times 4}$, and opacity values $\alpha^{v,m} \in \mathbb{R}^{H \times W}$, which are essential for novel view synthesis. Notably, the predicted pointmap serves as the center of the Gaussian, the input pixel color $I_v$ is used for its color and fix the spherical harmonics degree to be 0. During inference, A model with M branches is used but the final per-view Gaussian predictions are computed using the heads in the first branch.

## 3.2 Planar Geometry Formulation

To facilitate surface reconstruction from the predicted Gaussian parameters, we introduce a Depth-Normal Regularization strategy aimed at enhancing the accuracy of depth representation. This strategy leverages two fundamental planar geometric properties: normal and depth from our predicted 3D Gaussian primitives.

**Flattening 3D Gaussians.** To enhance the capacity of Gaussians in modeling surface geometry, we first apply a flattening operation to the Gaussian primitives. Inspired by Chen et al. (2023), we specifically introduce a scale regularization loss $\mathcal{L}_s$, which minimizes the smallest of the three scaling factors $\mathbf{S} = (s_1, s_2, s_3)^\top \in \mathbb{R}^3$ for each Gaussian:

$$\mathcal{L}_{\text{s}} = \|\min(s_1, s_2, s_3)\|_1. \tag{3}$$

By minimizing the loss, the Gaussian is driven towards a flat shape, effectively approximating a local surface plane.

**Normal Map Rendering.** Once a Gaussian is flattened onto a local plane, the surface normal $\mathbf{n}$ is computed from the predicted rotation quaternion $q$ and scaling factors $S$ predicted by our feedforward network. We first convert $q$ into a rotation matrix $R \in \mathbb{R}^{3 \times 3}$. The normal is then defined as the direction corresponding to the smallest scaling factor: $\mathbf{n} = R[k, :] \in \mathbb{R}^3, k = \text{argmin}([s_1, s_2, s_3])$. The normal $\mathbf{n}$ is subsequently transformed into the camera coordinate system. Finally, a rendered normal map $\hat{\mathbf{N}}$ is generated by a weighted summation of individual Gaussian normals $\mathbf{n}_i$ and their opacities $\alpha_i$ along each ray:

$$\hat{\mathbf{N}} = \sum_{i \in K} \mathbf{n}_i \alpha_i \prod_{j=1}^{i-1} (1 - \alpha_j) / \sum_{i \in K} \alpha_i \prod_{j=1}^{i-1} (1 - \alpha_j), \tag{4}$$

**Depth Map Rendering.** To achieve more precise and geometrically consistent depth values than simply using the Gaussian's center position Tang et al. (2024), we compute the depth as the intersection point of a viewing ray originating from the camera center with the plane represented by the flattened Gaussian. Formally, the intersection depth $\mathbf{d}(\mathbf{n}, \mathbf{p}, \mathbf{r})$ is calculated by:

$$\mathbf{d}(\mathbf{n}, \mathbf{p}, \mathbf{r}) = \mathbf{r}_z * (\mathbf{n} \cdot \mathbf{p}) / (\mathbf{n} \cdot \mathbf{r}), \tag{5}$$

where $\mathbf{r}_z$ is the z-value of the ray direction. This formulation reveals that the intersection depth of a Gaussian is jointly determined by its position and surface normal, thereby enabling a more geometrically grounded and accurate depth estimation. Leveraging this property, a view-consistent depth map $\hat{D}$ is rendered through a weighted summation of these intersection depths $d_i$, weighted by their opacities $\alpha_i$:

$$\hat{D} = \frac{\sum_{i \in K} d_i \alpha_i \prod_{j=1}^{i-1} (1 - \alpha_j)}{\sum_{i \in K} \alpha_i \prod_{j=1}^{i-1} (1 - \alpha_j)}, \tag{6}$$

where $K$ denotes the set of Gaussians along a ray, sorted by depth. The properties form the foundation for regularization.

### 3.3 GEOMETRY-AWARE LOSS FUNCTIONS

To train our model effectively, we employ a set of loss functions tailored to guide the learning of geometry-aware representations. We begin with a confidence-aware pointmap regression loss, denoted as $\mathcal{L}_{\text{conf}}$, which supervises the predicted 3D pointmaps $P^{k,m}$ using their associated confidence maps $Q^{k,m}$. The loss is defined as:

$$\mathcal{L}_c = \sum_{k,m} \sum_{p \in P^{k,m}} Q_p^{k,m} \left\| P_p^{k,m} - \mathbf{P}_{gt,p} \right\|_1 - \beta \log Q_p^{k,m} \tag{7}$$

where $P_p^{k,m}$ is the predicted 3D point, $\mathbf{P}_{gt,p}$ is the ground truth, $Q_p^{k,m}$ is the confidence score, and $\beta$ is a regularization parameter. We also employ a standard RGB rendering loss $\mathcal{L}_r$ Charatan et al. (2024a), which supervises the rendered image against the ground-truth RGB image to preserve photometric fidelity.

To improve surface regularity, we introduce a flattening loss $\mathcal{L}_s$, which encourages the predicted Gaussian primitives to lie on locally planar surfaces. In addition, we employ a rendered normal map loss $\mathcal{L}_n$ to align the rendered normal map $\hat{\mathbf{N}}$ with a reference normal map $\mathbf{N}_{\text{gt}}$, which is computed from the ground-truth depth via finite difference-based gradients. The loss combines an $\ell_1$ term and a cosine similarity term, and is defined as:

$$\mathcal{L}_n = |\hat{\mathbf{N}} - \mathbf{N}_{\text{gt}}|_1 + \left( 1 - \hat{\mathbf{N}} \cdot \mathbf{N}_{\text{gt}} \right), \tag{8}$$

where the first term enforces per-pixel accuracy, and the second term promotes angular alignment. This supervision encourages the rendered normals to more faithfully reflect the underlying scene geometry.

While explicit normal regularization can effectively refine the orientation of 3D Gaussians, it has less impact on their positions. To address this limitation and ensure robust 3D surface reconstruction, we introduce a Depth-Normal (D-Normal) regularization strategy Chen et al. (2024b), which enables joint optimization of both the orientation and positional accuracy of the Gaussians. The D-Normal $\overline{\mathbf{N}}_d$ is derived from the rendered depth $\hat{D}$ by computing the cross-product of horizontal and vertical finite differences from neighboring points:

$$\overline{\mathbf{N}}_d = \frac{\nabla_v \mathbf{d} \times \nabla_h \mathbf{d}}{|\nabla_v \mathbf{d} \times \nabla_h \mathbf{d}|}, \tag{9}$$

where $\mathbf{d}$ represents the 3D coordinates of a pixel obtained via back-projection from the depth map. Finally, the D-Normal regularization loss $\mathcal{L}_{dn}$ is defined as:

$$\mathcal{L}_{dn} = \left( \|\bar{\mathbf{N}}_d - \mathbf{N}_{gt}\|_1 + (1 - \bar{\mathbf{N}}_d \cdot \mathbf{N}_{gt}) \right), \tag{10}$$

**Overall Loss.** The final total loss $\mathcal{L}_{total}$ combines the geometric regularization losses for Gaussian primitives and the pointmap regression loss:

$$\mathcal{L}_{total} = \lambda_c \mathcal{L}_c + \lambda_r \mathcal{L}_r + \lambda_s \mathcal{L}_s + \lambda_n \mathcal{L}_n + \lambda_{dn} \mathcal{L}_{dn} \tag{11}$$

The weighting factors $\lambda_c$, $\lambda_r$, $\lambda_s$, $\lambda_n$, and $\lambda_{dm}$ balance the contributions of each loss term, ensuring a holistic optimization of the reconstructed geometry and pointmaps.

## 4 EXPERIMENTS

We begin by presenting the experimental setup in Sec. 4.1. We assess the effectiveness and generalization capability of our approach for surface reconstruction in Sec. 4.2. We further demonstrates the novel view synthesis capability of our method in Sec. 4.3. Additionally, we validate the effectiveness of the proposed techniques in Sec. 4.4.

### 4.1 IMPLEMENTATION DETAILS

We train our model on ScanNet++ dataset Yeshwanth et al. (2023). View sequences $\{I_v\}_{v=1}^N$ are generated with an overlap–based sampler. Starting from a random keyframe, a candidate view is appended whenever the overlap between its point cloud and the accumulated scene cloud falls within

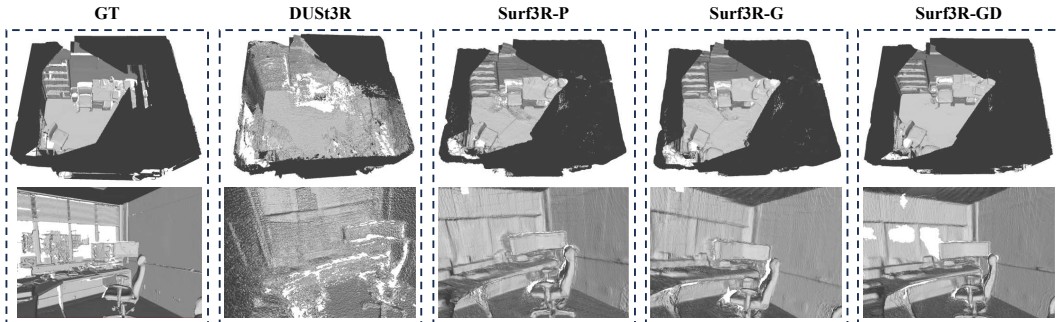

Figure 3: Qualitative comparison of surface reconstruction results on ScanNet++ dataset.

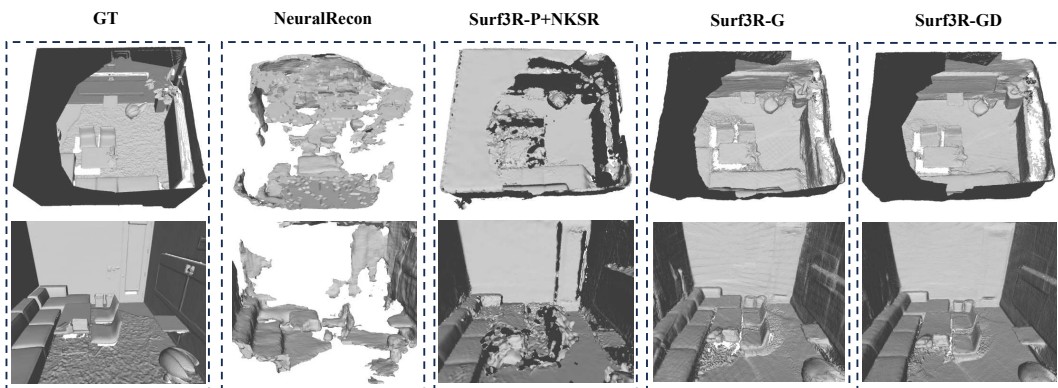

Figure 4: Qualitative comparison of zero-shot surface reconstruction results on Replica dataset.

30%–70%. For training, each scene provides 100 trajectories of 10 views, yielding diverse yet geometrically consistent inputs. For validation, we construct 1 000 trajectories with 30 views on the ScanNet++ validation split and retain the 50 widest-baseline views per scene for surface fusion to maximise spatial coverage. Additionally, we conduct zero-shot generalization experiments on Replica dataset Straub et al. (2019) to assess the cross-dataset adaptability of our model.

We train using 32 NVIDIA H800 GPUs, processing input views at a resolution of $224 \times 224$. For each training trajectory, the first $N = 8$ views are used as input, from which $M = 4$ reference views are randomly selected. The model is trained for 50 epochs, resulting in a total training time of approximately 40 hours. Additional training details are provided in the Appendix.

## 4.2 SURFACE RECONSTRUCTION

We evaluate three progressively enhanced variants of our framework. **Surf3R-P** employs only the pointmap heads and is trained with the reconstruction loss $\mathcal{L}_c$. **Surf3R-G** extends this baseline by introducing Gaussian heads together with the associated rendering loss $\mathcal{L}_r$. **Surf3R-GD**, the full approach, further incorporates the D-Normal regularization strategy and is optimized with the additional terms $\mathcal{L}_s$, $\mathcal{L}_n$ and $\mathcal{L}_{dn}$. As shown in Tab. 1, the results for per-scene methods are computed on the eight ScanNet++ validation scenes, while feed-forward models are evaluated across all 50 scenes, with the table reporting the dataset-wide averages. Surf3R-GD achieves state-of-the-art surface reconstruction performance, with an F1-score of 78.71. Compared to traditional per-scene reconstruction approaches such as NeuS Wang et al. (2021a), 2DGS Huang et al. (2024), SuGaR Guédon & Lepetit (2024), and PGSR Chen et al. (2024a), our method achieves significantly higher surface reconstruction quality. In particular, compared to the concurrent method SuGaR Guédon & Lepetit (2024), our approach yields a substantial improvement (78.71 vs. 36.12 in F1-score). Moreover, our model exhibits exceptional efficiency, offering a reconstruction speed that is approximately $180\times$ faster than per-scene methods. We further compare our method with feedforward-based approach DUSt3R Wang et al. (2024). DUSt3R and Surf3R-P reconstruct surfaces by first back-projecting point clouds to depth maps, which are then fused via TSDF. In contrast, both Surf3R-G and Surf3R-

| Method | Per-Scene | | | | Feedforward | | | |
|---|---|---|---|---|---|---|---|---|
| | NeuS | 2DGS | SuGaR | PGSR | DUSt3R | Surf3R-P | Surf3R-G | Surf3R-GD (Ours) |
| Precision ↑ | 29.42 | 23.01 | 38.30 | 35.33 | 4.62 | 63.75 | 78.50 | **80.24** |
| Recall ↑ | 22.14 | 16.04 | 34.92 | 21.70 | 4.84 | 62.36 | 75.34 | **77.55** |
| F1-score ↑ | 25.13 | 18.30 | 36.12 | 24.92 | 4.06 | 62.89 | 76.72 | **78.71** |
| Time | > 30 min | | | | > 1 min | < 10 s | | |

Table 1: **Quantitative comparison on ScanNet++ dataset. Bold** indicates best result. Our method achieves state-of-the-art performance across all metrics. *Surf3R-P*: point-map heads, trained with $\mathcal{L}_c$; *Surf3R-G*: + Gaussian heads, adds $\mathcal{L}_r$; *Surf3R-GD* (Full model): + D-Normal regularization, adds $\mathcal{L}_s$, $\mathcal{L}_n$ and $\mathcal{L}_{dn}$.

GD leverage Gaussian rendering to directly estimate high-quality depth maps for mesh reconstruction. As shown in Tab. 1, Surf3R-P achieves a significant improvement over DUSt3R, which requires explicit global alignment, with an F1-score increasing from 4.06 to 62.89. This underscores the advantage of aggregating geometric cues across all input views rather than relying on pairwise stereo matches processed one at a time. Moreover, enriching Surf3R-P with a 3D Gaussian representation (Surf3R-G) enhances global geometric consistency and raises the F1-score from 62.89 to 76.72. And the additional introduction of D-Normal regularization (Surf3R-GD) pushes it further to 78.71, yielding the best overall performance. As shown in Fig 3, our approach yields more accurate and complete reconstructions, particularly excelling at recovering planar surfaces and capturing fine-grained geometric details.

Moreover, our method demonstrates strong generalization capabilities. As shown in Tab. 2, under zero-shot inference on the Replica dataset, Surf3R-GD also achieves state-of-the-art performance with an F1-score of 41.92. Compared to traditional methods such as NeuralRecon Sun et al. (2021), DUSt3R Wang et al. (2024), and Surf3R-P+NKSR Huang et al. (2023), it consistently outperforms all baselines. As shown in Fig. 4, our method produces more complete and faithful surfaces, highlighting the superior generalization capability of our approach under unseen scenes.

| Method | NeuralRecon | DUSt3R | Surf3R-P | Surf3R-P + NKSR | Surf3R-G | Surf3R-GD (Ours) |
|---|---|---|---|---|---|---|
| Precision ↑ | 14.61 | 20.16 | 22.31 | 24.14 | 24.86 | **36.66** |
| Recall ↑ | 12.33 | 14.80 | 21.52 | 31.37 | 32.06 | **49.04** |
| F1-score ↑ | 13.41 | 16.90 | 21.88 | 27.16 | 27.96 | **41.92** |

Table 2: **Quantitative comparison on Replica dataset. Bold** indicates best result. Our method achieves superior performance across all metrics. *Surf3R-P*: point-map heads, trained with $\mathcal{L}_c$; *Surf3R-G*: + Gaussian heads, adds $\mathcal{L}_r$; *Surf3R-GD* (Full model): + D-Normal regularization, adds $\mathcal{L}_s$, $\mathcal{L}_n$ and $\mathcal{L}_{dn}$.

### 4.3 MULITI-VIEW NVS ON SCANNET++

As shown in Tab. 3, Surf3R-GD consistently achieves the best novel view synthesis performance across all multi-view configurations on the ScanNet++ dataset. With only 4 input views, it outperforms all baselines, achieving a PSNR of 15.06, SSIM of 0.66, and LPIPS of 0.26, demonstrating robust geometry-aware synthesis even under sparse view conditions. As the number of input views increases to 12 and 24, our method maintains leading performance, particularly in perceptual quality metrics. Notably, the LPIPS score drops to 0.23 at 24 views, outperforming DUSt3R (0.68), indicating more stable view synthesis and sharper geometric details. These results underscore the strong generalization capability of our geometry-guided surface reconstruction framework, which not only delivers accurate 3D geometry but also enables high-quality NVS across varying input densities.

### 4.4 ABLATION STUDY

**View ablations.** To investigate the impact of input view count on reconstruction quality, we conduct an ablation study by varying the number of input views during inference. As shown in Table 4, our method achieves the best performance when using 50 input views, with an F1-score of 41.92. Interestingly, increasing the number of views beyond this point does not lead to further improvements

|  | 4 Views | | | 12 Views | | | 24 Views | | |
|---|---|---|---|---|---|---|---|---|---|
|  | PSNR ↑ | SSIM ↑ | LPIPS ↓ | PSNR ↑ | SSIM ↑ | LPIPS ↓ | PSNR ↑ | SSIM ↑ | LPIPS ↓ |
| DUSt3R | 11.66 | 0.47 | 0.63 | 10.72 | 0.46 | 0.67 | 10.81 | 0.40 | 0.68 |
| Surf3R-G | 13.21 | 0.64 | 0.31 | 16.77 | 0.55 | 0.30 | 17.69 | 0.55 | 0.26 |
| Surf3R-GD | **15.06** | **0.66** | **0.26** | **17.72** | **0.61** | **0.24** | **18.08** | **0.58** | **0.23** |

Table 3: **NVS results on ScanNet++ dataset. Bold** indicates best result. Our method achieves NVS rendering quality comparable with other Gaussian-based methods. *Surf3R-G*: + Gaussian heads, adds $\mathcal{L}_r$; *Surf3R-GD* (Full model): + D-Normal regularization, adds $\mathcal{L}_s$, $\mathcal{L}_n$ and $\mathcal{L}_{dn}$.

| Views | 10 Views | 30 Views | 50 Views | 70 Views | 100 Views |
|---|---|---|---|---|---|
| Precision ↑ | 9.84 | 18.41 | **36.66** | 35.42 | 34.87 |
| Recall ↑ | 17.57 | 30.38 | **49.04** | 47.24 | 45.29 |
| F1-score ↑ | 12.29 | 22.88 | **41.92** | 40.47 | 39.37 |

Table 4: **View Ablation Study on Replica dataset. Bold** indicates best result. Performance comparison under different numbers of input views.

and may even slightly degrade performance. We attribute this to accumulated pose estimation errors from the input point clouds, which become more pronounced as the number of views increases, ultimately affecting the mesh reconstruction quality.

**Branch and Loss ablations.** As shown in Tab. 5, when restricting the network to a single branch (one reference view) degrades the F1-score from 36.66 to 23.24 (Column A). This indicates that, with sparse views, a single reference cannot establish reliable geometric correspondences across wide baselines, resulting in poor reconstructions. We also verify the effectiveness of different regularization terms on reconstruction quality. As shown in Tab. 5, both components contribute significantly to the reconstruction quality. Excluding the scale term (Column B) and the normal term (Column C) results in a notable decline in all metrics. Notabaly, removing the D-Normal term (Column D) leads to a substantial drop in performance, with the F1-score decreasing from 41.92 to 30.96. This suggests that the D-Normal regularization plays a critical role in encouraging the predicted normals to align with the underlying surface geometry. Our full model (Column E) achieves the best performance across all metrics, demonstrating the benefits of both components in producing surface reconstructions.

| Metric | A. w/o Multi-branch | B. w/o Scale | C. w/o Normal | D. w/o D-Normal | E. Full |
|---|---|---|---|---|---|
| **Precision** ↑ | 23.24 | 32.9 | 34.5 | 25.38 | **36.66** |
| **Recall** ↑ | 30.90 | 43.20 | 45.80 | 39.69 | **49.04** |
| **F-score** ↑ | 26.53 | 37.35 | 39.35 | 30.96 | **41.92** |

Table 5: **Branch and Loss Ablation Study on Replica dataset**. **Bold** indicates best result. Performance with different regularization terms.

## 5 CONCLUSION

In this paper, we propose Surf3R, a novel feed-forward framework for pose-free 3D surface reconstruction from sparse multi-view RGB inputs. Unlike traditional MVS methods that rely heavily on accurate camera calibration and iterative alignment, Surf3R eliminates the need for camera intrinsics or extrinsics by leveraging a cross-view attention mechanism and a multi-branch cross-reference fusion strategy. This enables effective feature propagation across arbitrarily selected views and mitigates the degradation caused by large viewpoint gaps. Additionally, We introduce a novel Depth-Normal Regularizer grounded in 3D Gaussian representations, which integrates normal estimation into the geometric parameter learning process, yielding more consistent and detailed surfaces. Extensive experiments on benchmark datasets such as ScanNet++ and Replica demonstrate that Surf3R achieves state-of-the-art performance in surface reconstruction while maintaining strong generalization ability in unseen scenarios.

# Ethics Statement

This work introduces Surf3R, a novel feed-forward approach for pose-free 3D surface reconstruction from sparse RGB views. The research is foundational and focused on algorithmic development, with no involvement of human subjects, personal data, or social risk assessment. All experiments were conducted using publicly available benchmark datasets (e.g., ScanNet++, Replica), following established academic practices. We advocate for the responsible use of this technology, ensuring compliance with ethical guidelines and legal standards. The authors declare no conflicts of interest.

# Reproducibility Statement

To ensure the reproducibility of our results, we provide full details of the Surf3R framework and its implementation. The method, including the multi-branch architecture, cross-reference fusion, and D-Normal regularization, is comprehensively described in Section 3. Additionally, all experimental setups, such as network architectures, hyperparameters, and optimization settings, are outlined in Appendix A.

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

# A APPENDIX

## A.1 EXPERIMENTAL SETTINGS

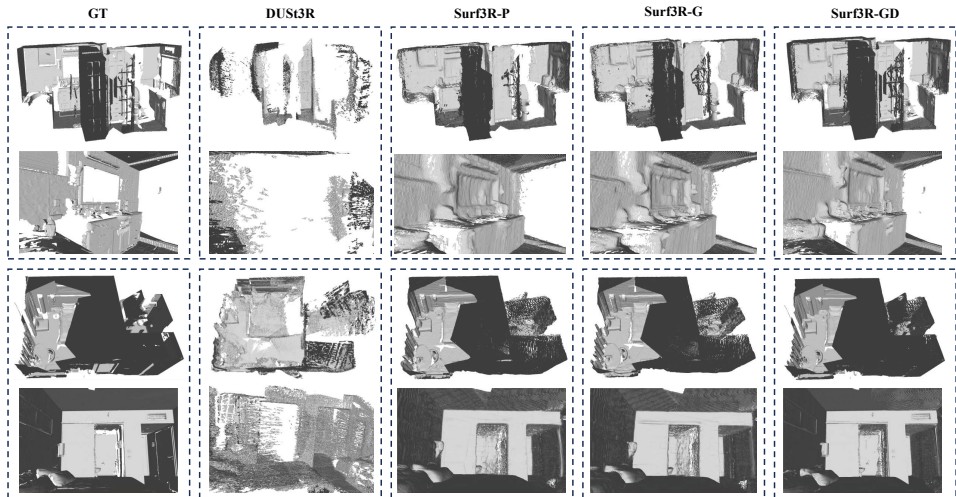

Figure A: More qualitative comparisons of surface reconstruction results on ScanNet++ dataset.

**Training Recipe.** Our method is implemented in the PyTorch framework Paszke et al. (2019). Each model is optimized for 50 epochs with the Adam optimizer Diederik (2014), starting from an initial learning rate of 1.5e-4 that follows a cosine decay schedule. Training takes 40 hours. The hyperparameters in the loss functions are $\lambda_c = 1$, $\lambda_r = 1$, $\lambda_s = 10$, $\lambda_n = 0.5$, and $\lambda_{dn} = 0.5$. The input image resolution is $224 \times 224$. In our experiments, we use Scannet++ Yeshwanth et al. (2023) (856 scenes for train, 50 scenes for test). To further assess the generalization capability of our method, we conduct a zero-shot evaluation on the Replica dataset Straub et al. (2019).

**Training Trajectoties.** Trajectory generation begins by randomly selecting one of the predefined views in the ScanNet++ dataset to serve as the initial viewpoint. Each chosen viewpoint yields an associated point map: for a view indexed by $I^i$, we denote its point map by $X^i$. The set $X^j{}_{j<i}$ comprises the point maps of all previously selected views. Using this set, we compute the overlap ratio $O(X^i, X^j)$ for every pair $(X^i, X^j)$. Formally,

$$O(X^i, X^j) = \frac{1}{2}\Big(\text{Cov}(X^i, X^j) + \text{Cov}(X^j, X^i)\Big), \tag{1}$$

$$\text{Cov}(X^i, X^j) = \frac{1}{|A|} \sum_{p \in X^i} \big[\text{NearestDis}(p, X^j) < t_c\big], \tag{2}$$

Here, $|A|$ denotes the number of points in $X^i$, and the distance threshold is fixed at $t_c = 0.0015$. A candidate view $X^i$ is accepted only when its maximum overlap with any previously selected map falls within the interval $[t_{\min,}, t_{\max}]$, i.e.,

$$\max_j O(X^i, X^j) \in [\,t_{\min}, t_{\max}\,] \tag{3}$$

where we set $t_{\min} = 30\%$ and $t_{\max} = 70\%$.

For evaluation we generate 1 000 trajectories using the procedure described above, with one modification. Each trajectory contains 30 camera views: the first 24 serve as conditioning inputs, whereas the remaining 6 are withheld as novel targets for neural view synthesis (NVS). The $k$-th novel view ($k = 1, \ldots, 6$) is chosen so that it is well covered by conditioning views $4k - 3$ through $4k$. When evaluating NVS with $m$ input views, we employ the first $\lceil m/4 \rceil$ novel targets. For example, the first novel view when $m = 4$, the first two when $m = 8$, and so on.

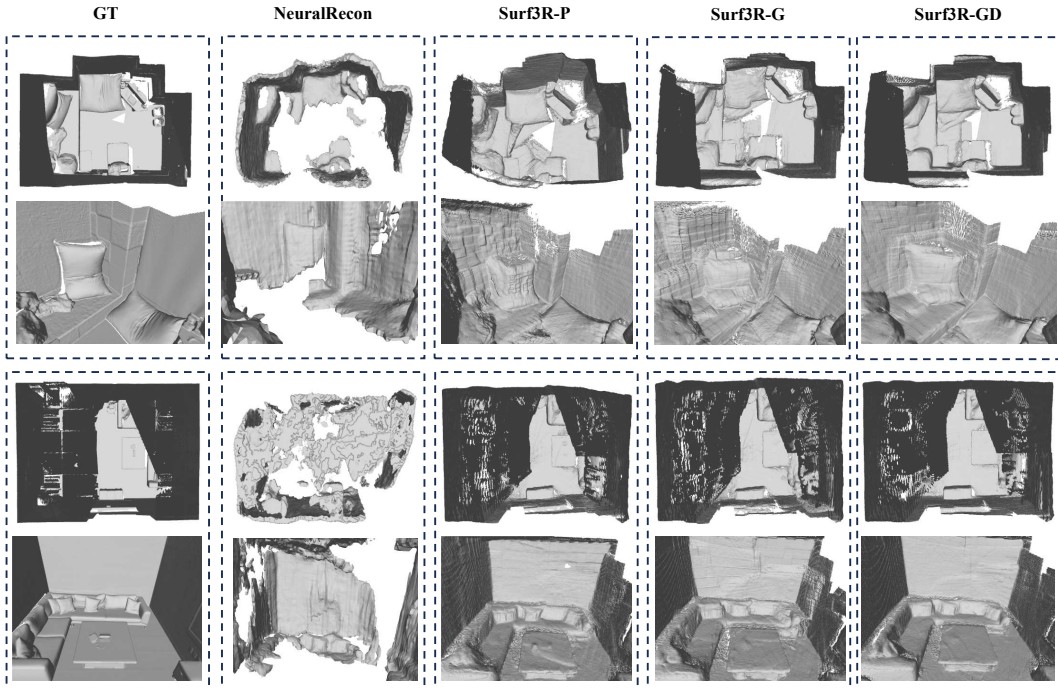

| GT | NeuralRecon | Surf3R-P | Surf3R-G | Surf3R-GD |

Figure B: More qualitative comparisons of zero-shot surface reconstruction results on Replica dataset.

**Surface Reconstruction.** Following prior works Chen et al. (2024b;a), we report percision, recall, and F1-score. The mesh extraction method used is consistent with 2DGS Huang et al. (2024) . Given rendered depth maps and camera poses, these inputs are fused via Open3D's TSDF integration to construct a continuous Signed Distance Field (SDF). The final surface mesh is then directly extracted from the SDF at its zero-level isosurface using Marching Cubes. enabling direct geometry reconstruction without intermediate point cloud representations.

**Novel View Synthesis.** Consistent with recent studies Smart et al. (2024); Charatan et al. (2024b), we assess image quality using three standard metrics: Peak Signal-to-Noise Ratio (PSNR), Structural Similarity Index Measure (SSIM), and Learned Perceptual Image Patch Similarity (LPIPS). We compare with a DUSt3R-based baseline, which generates per-pixel Gaussian parameters as follows. We use the pointmap predicted by DUSt3R as the Gaussian center, use pixel RGB color as the color, a constant 0.001 for the scale factor, an identity transform, 1.0 for opacity, and spherical harmonics with zero-degree.

## A.2 MORE EXPERIMENTS

In the original ScanNet++ benchmark, the per-scene baseline could be evaluated on only eight scenes because of its prohibitive runtime. To ensure a fair comparison, we therefore assess feedforward approach on the same eight scenes, the results are reported in Tab. A. On these eight scenes, our method consistently outperforms the original per-scene approach, further demonstrating the effectiveness of our approach.

## A.3 VISUALIZATION

To further demonstrate the effectiveness of our method, we provide additional visual results that highlight the quality of the surface reconstruction. We present additional surface reconstruction results on the ScanNet++ dataset, as shown in Fig. A, which demonstrate the strong surface reconstruction capability of our method. Similarly, as illustrated in Fig. B, our method exhibits excellent generalization performance on the Replica dataset.

| Method | DUSt3R | Surf3R-P | Surf3R-G | Surf3R-GD (Ours) |
|---|---|---|---|---|
| Precision ↑ | 4.82 | 57.04 | 76.25 | **80.00** |
| Recall ↑ | 3.73 | 58.10 | 74.62 | **77.53** |
| F1-score ↑ | 4.01 | 57.22 | 75.43 | **78.27** |
| *Time* | > 1 min | | *< 10 s* | |

Table A: **Quantitative comparison on 8 scenes of ScanNet++ dataset. Bold** indicates best result. Our method achieves state-of-the-art performance across all metrics. *Surf3R-P*: point-map heads, trained with $\mathcal{L}_c$; *Surf3R-G*: + Gaussian heads, adds $\mathcal{L}_r$; *Surf3R-GD* (Full model): + D-Normal regularization, adds $\mathcal{L}_s$, $\mathcal{L}_n$ and $\mathcal{L}_{dn}$.

## A.4 MORE ABLATIONS

**Gaussian abalations.** To evaluate the effectiveness of our Gaussian-based depth rendering, we further conduct a comparison with the point-to-depth approach. Specifically, the point cloud, generated by Surf3R-GD, is back-projected onto the image plane to produce depth maps. TSDF fusion is then applied to these depth maps to obtain the final meshes. As shown in Table B, this baseline achieves significantly lower performance, with an F1-score of only 34.41. In contrast, our Gaussian rendering strategy achieves a much higher F1-score of 41.92. The results validate that explicitly modeling uncertainty and density through Gaussians leads to more faithful geometry compared to sparse and noisy point cloud projections.

| Ablation Item | Precision ↑ | Recall ↑ | F1-score ↑ |
|---|---|---|---|
| A. point to depth | 30.63 | 39.29 | 34.41 |
| B. gaussian to depth | 36.66 | 49.04 | 41.92 |

Table B: **Gaussian Ablation on Replica dataset**. Comparison with the point-to-depth baseline.

## A.5 USE OF LARGE LANGUAGE MODELS

We used large language models (LLMs) solely to assist in polishing the writing (e.g., grammar, wording, and minor LaTeX fixes). No algorithms, analyses, results, or claims were generated by LLMs; all technical content and decisions were made by the authors. LLMs were not involved in creating or labeling data, and no evaluation items were exposed.

## A.6 LIMITATIONS

Although our method achieves impressive surface reconstruction results and reconstructs scenes at a very fast speed, we have observed that its performance on transparent objects is less satisfactory. Specifically, as shown in Fig. A, the reconstruction of glass surfaces is not ideal. This issue stems from the point cloud generated by Surf3R-GD. In the future, we aim to address this limitation and further improve the precision of surface reconstruction.

