# OpenReview forum: "Surf3R: Rapid Surface Reconstruction from Sparse RGB Views in Seconds"
_ICLR.cc/2026/Conference — Submitted to ICLR 2026_

### Official Review · Reviewer_oKpa · 2025-10-29

**Soundness:** 2
**Presentation:** 2
**Contribution:** 2
**Rating:** 2
**Confidence:** 4

**Summary:**

Surf3R proposes a feedforward network for efficient surface reconstruction from uncalibrated RGB images. It uses a multi-branch design to process multiple reference views, capturing complementary geometric information. The method employs 3D Gaussian splatting for surface representation, allowing smooth and continuous reconstructions. To improve accuracy, a Depth-Normal (DN) regularization is introduced, optimizing both surface depth and normal consistency. Feature-Refine (FR) blocks and Cross-Reference Fusion (CRF) blocks are used to refine multi-view features and aggregate information across views. Surf3R is trained with geometry-aware losses, achieving fast real-time reconstructions in under 10 seconds while maintaining strong performance across different datasets.

**Strengths:**

- Successfully tackles the underexplored problem of pose-free surface reconstruction from uncalibrated cameras, removing the need for prior camera calibration.
- Achieves real-time surface reconstruction in under 10 seconds, making it suitable for practical, interactive applications.

**Weaknesses:**

1. **Writing and Presentation:** The citation formatting is frequently inconsistent or incorrect, which hinders readability.
2. **Experimental Rigor and Clarity:** The validation of results is not sufficiently clear. Specifically, the comparison to Dust3R lacks detail, raising concerns about the fairness and validity of the evaluation.
3. **Missing Baselines (Reconstruction):** The experimental comparison could be strengthened by including other recent, high-performant baselines for feed-forward, pose-free point cloud reconstruction (e.g., Mast3R[1], VGGT[2]).
4. **Missing Baselines (GS):** For the meshing results (Surf3R-G and Surf3R-GD), the paper would be more comprehensive if it included comparisons against other relevant pose-free Gaussian splatting (e.g., PFSplat[3], NoPoSplat[4]) combined with meshing (e.g. TSDF).
5. **Novelty and Justification:** The method's conceptual novelty appears limited, as several components (e.g., the D-Normal loss) are incremental adaptations of existing work [5]. The core novel components, such as the multi-reference view structure of the network, currently lack sufficient theoretical motivation and experimental validation.

---
[1] Leroy, Vincent, Yohann Cabon, and Jérôme Revaud. "Grounding image matching in 3d with mast3r." European Conference on Computer Vision. Cham: Springer Nature Switzerland, 2024.

[2] Wang, Jianyuan, et al. "Vggt: Visual geometry grounded transformer." Proceedings of the Computer Vision and Pattern Recognition Conference. 2025.

[3] Hong, Sunghwan, et al. "PF3plat: Pose-Free Feed-Forward 3D Gaussian Splatting for Novel View Synthesis." Forty-second International Conference on Machine Learning.

[4] Ye, Botao, et al. "No Pose, No Problem: Surprisingly Simple 3D Gaussian Splats from Sparse Unposed Images." The Thirteenth International Conference on Learning Representations.

[5] Chen, Hanlin, et al. "Vcr-gaus: View consistent depth-normal regularizer for gaussian surface reconstruction." Advances in Neural Information Processing Systems 37 (2024): 139725-139750.

**Questions:**

1. **Dust3R Comparison Protocol:** Regarding the Dust3R[1] baseline, please clarify the experimental protocol:
    - How was rendering quality evaluated against Dust3R, given its output is a colored point cloud?
    - How was scale ambiguity handled? Please specify the evaluation domain (e.g., normalized or original reconstructed domain) used for this comparison.
2. **Justification of Novel Components:** Following Weakness #5, please provide a deeper analysis or ablation study to justify the specific design choices for the multi-reference view structure and the partitioned head design.
3. **Generalizability:** To better demonstrate the method's generalizability, we request an evaluation on other standard multi-view datasets, such as DTU[2] or BlendedMVS[3].

---
[1] Wang, Shuzhe, et al. "Dust3r: Geometric 3d vision made easy." Proceedings of the IEEE/CVF Conference on Computer Vision and Pattern Recognition. 2024.

[2] Jensen, Rasmus, et al. "Large scale multi-view stereopsis evaluation." Proceedings of the IEEE conference on computer vision and pattern recognition. 2014.

[3] Yao, Yao, et al. "Blendedmvs: A large-scale dataset for generalized multi-view stereo networks." Proceedings of the IEEE/CVF conference on computer vision and pattern recognition. 2020.

---

### Official Review · Reviewer_Rp98 · 2025-10-30

**Soundness:** 2
**Presentation:** 2
**Contribution:** 2
**Rating:** 2
**Confidence:** 4

**Summary:**

This paper proposes an end-to-end feedforward approach that reconstructs 3D surfaces from sparse views without camera calibration.

D-Normal regularization is proposed for surface reconstruction based on an explicit 3D Gaussian representation.

Experiments demonstrate superior results compared to other baselines like neuralrecon and dust3r, and excellent efficiency.

**Strengths:**

1. The writing is easy to follow
2. The proposed pipeline is efficient and achieves superior results compared to Dust3R and NeuralRecon

**Weaknesses:**

Major:
1. The paper claims to propose a feed-forward approach for 3D surface reconstruction without camera calibration as one of its main contributions. However, this idea appears conceptually similar to Dust3R [1], VGGT [2], and their subsequent works. Moreover, the proposed Depth–Normal Regularization seems to originate from another existing paper [Chen et al. (2024b)]. Therefore, the novelty of this work is unclear. I encourage the authors to clarify how their approach differs fundamentally from these prior methods in the rebuttal.

2. It is not clearly stated what coordinate system the predicted point clouds are represented in. Are they in the camera coordinates of the reference view? In addition, how does the proposed method estimate camera poses from its predictions?

3. Several important baselines are missing. In particular, more recent works following DUSt3R, such as Cut3R [3], VGGT [2], and SLAM-based approaches like WildGS-SLAM [4], should be included for a fair comparison. The authors should explain why these methods were not considered as baselines in the current evaluation.

Minor:

1. Lines 147–150 state that the primary focus of VGGT lies in view synthesis rather than high-fidelity surface reconstruction. I do not think this statement is accurate, as VGGT has demonstrated strong performance in reconstruction-related tasks.

2. For the novel view synthesis (NVS) evaluation, it would strengthen the paper to include comparisons with more recent state-of-the-art NVS methods rather than only Dust3R.

[1] Wang, Shuzhe, et al. "Dust3r: Geometric 3d vision made easy." Proceedings of the IEEE/CVF Conference on Computer Vision and Pattern Recognition. 2024.
[2] Wang, Jianyuan, et al. "Vggt: Visual geometry grounded transformer." Proceedings of the Computer Vision and Pattern Recognition Conference. 2025.
[3] Wang, Qianqian, et al. "Continuous 3d perception model with persistent state." Proceedings of the Computer Vision and Pattern Recognition Conference. 2025.
[4] Zheng, Jianhao, et al. "Wildgs-slam: Monocular gaussian splatting slam in dynamic environments." Proceedings of the Computer Vision and Pattern Recognition Conference. 2025.

**Questions:**

I am curious why the proposed cross-attention and multi-branch cross-reference fusion mechanisms outperform the local and global self-attention used in the VGGT paper. It would be helpful if the authors could provide a more detailed explanation or ablations demonstrating the advantage of their design.

In addition, I suggest that the paper more clearly differentiate the proposed approach from existing baselines such as DUSt3R, VGGT, and their follow-up works, emphasizing what specific innovations or insights this method introduces beyond them.

---

### Official Review · Reviewer_ufbt · 2025-10-31

**Soundness:** 2
**Presentation:** 2
**Contribution:** 1
**Rating:** 0
**Confidence:** 4

**Summary:**

The paper presents Surf3R, a method for feed-forward 3D surface reconstruction from unposed sparse views. Following the previous work of DUSt3R, the authors propose to use a multi-view, transformer-based architecture to 3D Gaussians instead of point maps that are then supervised via previously introduced confidence-aware point map regression, RGB rendering, normal, and 3D Gaussian flattening losses.
Additionally, the paper employs a D-Normal regularization loss, also introduced in prior work (VCR-GauS, NeurIPS 2024).
Experiments on ScanNet++ and zero-shot evaluation on Replica show improvements over feed-forward and optimization-based baselines.
The authors further ablate on the use of 3D Gaussians instead of point maps and TSDF fusion as well as the use of the D-Normal regularization loss.

**Strengths:**

- The paper addresses a challenging and interesting problem: Feed-forward 3D surface reconstruction from sparse and unposed RGB images.
  - To the best of my knowledge, it is the first work in the line of DUSt3R follow-ups that focuses on direct watertight surface reconstruction.
- The qualitative and quantitative evaluation shows advantages over the chosen baselines.
- The ablation studies regarding use of 3D Gaussians instead of just point maps and the use of the Depth-Normal regularization strategy from prior work validate the effectiveness of the design choices.
- The paper and appendix provide details about the training recipe, helpful for reproducibility of experiments.

**Weaknesses:**

- The authors blatantly sell ideas of two existing papers as their own contributions, while citing only one of these two papers insufficiently and without clearly stating what is their contribution and what not:
  - The architecture is 1:1 copied from MV-DUSt3R [1] without any citation of this work in this paper.
    - This paper proposes Feature-Refine (FR) blocks that correspond to the DecBlocks in MV-DUSt3R and the paper also shares almost the same notation (see equation 1 this paper vs equation 1 in MV-DUSt3R).
    - This paper proposes Cross-Reference Fusion (CRF) blocks that correspond to the CrossRefViewBlock in MV-DUSt3R.
    - Furthermore, in MV-DUSt3R, they have an additional MV-DUSt3R+ version for novel view synthesis by adding a 3D Gaussian head, which this paper does as well.
    - The method figure 2 of this paper is quite similar to Fig. 3 and Fig. 5 of MV-DUSt3R.
  - The Depth-Normal Regularization is adopted from VCR-GauS [2], but sold as one of this paper's contributions.
    - The authors insufficiently reference this paper twice in lines 86 and 295: " we introduce a Depth-Normal Regularization strategy Chen et al. (2024b)" " we introduce a Depth-Normal (D-Normal) regularization strategy Chen et al. (2024b)".
    - They never state anything regarding what this prior work has done.
    - This paper includes an entire section of replicated content without any citation to this paper. Section 3.2 (Planar Geometry Formulation) in this paper clearly reformulates parts of Sections 3.2 and Sections 3.3 in VCR-GauS without citing it.
    - The Depth-Normal Regularization strategy that is copied from VCR-GauS is also mentioned without citation in their contributions in the introduction (line 104): "Furthermore, we introduce a Depth-Normal Regularization strategy to enhance geometric fidelity."
- The related works and baselines are outdated.
  - Regarding feed-forward 3D reconstruction from sparse and unposed images, the paper mostly limits itself to DUSt3R as reference, ignoring more recent follow-ups like MV-DUSt3R [1], VGGT [3], MASt3R-SfM [4], except for one sentence in related work, citing some of them (lines 147ff.).
  - Regarding optimization-based surface reconstruction approaches, the paper is also not up-to-date, missing Spurfies [5] and MAtCha Gaussians [6] in both related work and baselines and even missing VCR-GauS [2] as baseline, from which the Depth-Normal Regularization strategy was copied.
- Despite adopted from MV-DUSt3R, the architecture description in the method section is difficult to understand.

References:
- [1] MV-DUSt3R+: Single-Stage Scene Reconstruction from Sparse Views In 2 Seconds. CVPR 2025
- [2] VCR-GauS: View Consistent Depth-Normal Regularizer for Gaussian Surface Reconstruction. NeurIPS 2024
- [3] VGGT: Visual Geometry Grounded Transformer. CVPR 2025
- [4] MASt3R-SfM: a Fully-Integrated Solution for Unconstrained Structure-from-Motion. 3DV 2025
- [5] Spurfies: Sparse Surface Reconstruction using Local Geometry Priors. 3DV 2025
- [6] MAtCha Gaussians: Atlas of Charts for High-Quality Geometry and Photorealism From Sparse Views. CVPR 2025

**Questions:**

Since my main concern is the copy of existing papers as own contributions, my only question to the authors is whether there is any other explanation for what I described in the weaknesses section and whether I missed anything.

**Details Of Ethics Concerns:**

I am quite confident that this is a case of plagiarism.
I already commented to the AC before submitting the review regarding the comparison with VCR-GauS (NeurIPS 2024) and how this paper tries to sell ideas from that paper as their own contributions, reformulating entire paragraphs, and missing references at important points in the paper (Section 3.2).
During further review of the paper, I noticed that the architecture up to notation is copied from MV-DUSt3R (CVPR 2025) without any reference of this paper at all.
Please see my weaknesses section for further details.

---

### Official Review · Reviewer_BumT · 2025-11-02

**Soundness:** 3
**Presentation:** 3
**Contribution:** 2
**Rating:** 4
**Confidence:** 4

**Summary:**

The paper proposes a feed-forward, pose-free surface reconstruction method that recovers 3D surfaces from sparse RGB inputs without requiring camera calibration or pose estimation. Unlike traditional Structure-from-Motion (SfM) or Multi-View Stereo (MVS) pipelines, which rely on heavy preprocessing, the proposed approach enables fast surface reconstruction through a pose-free feed-forward model. The key component is a multi-branch, multi-view decoder that jointly processes multiple reference views using Feature-Refine (FR) and Cross-Reference Fusion (CRF) blocks to facilitate long-range cross-view interaction. Additionally, a depth-normal regularizer based on 3D Gaussian representation is introduced to enhance 3D consistency and surface detail. Experiments on ScanNet++ and Replica datasets demonstrate that the proposed method can achieve good surface reconstruction quality compared to baselines, while also supporting novel view synthesis.

**Strengths:**

* The paper addresses a highly relevant problem. Achieving end-to-end 3D reconstruction without camera pose estimation is significant, as traditional pipelines that rely on Structure-from-Motion (SfM) for pose estimation are computationally expensive and often fragile under sparse inputs.

* The motivation of the paper is clear, and the paper is well-written and the proposed method is easy to follow and the overall presentation is clear.

* Experimental results show that this method can achieve high quality surface reconstruction (Table 1, 2 and Fig. 3 & 4) compared to the baselines on both within dataset and cross dataset.

**Weaknesses:**

* While the proposed multi-branch design and cross-reference fusion blocks appear effective for large-scale scene-level reconstruction from sparse views, the overall architectural concept feels relatively straightforward. Similar design principles though applied in different contexts have been explored in prior works [1,2].

* The combination of FR and CRF blocks, built on a multi-branch design, appears effective for representing scenes and facilitating feature communication across views. However, it is unclear how scalable this design is, as the multiple attention modules across branches likely make it memory- and computation-intensive. A discussion or experiment analyzing the computational and memory overhead of this design would greatly strengthen the paper.

* Although the method achieves good results over the point map based baselines like DusTr, I feel the method should also compare against other sparse view reconstruction methods based on 3DGS like PixelSplat [3] or HiSplat [4]. A comparison with these methods, will help to further reinforce the claims of the paper.

* A more comprehensive discussion on the failure mode will be helpful; currently it only describes its poor performance for transparent objects which is not interesting.

**Questions:**

Please refer to weakness.

[1] Tang, Shitao, et al. "Mvdiffusion++: A dense high-resolution multi-view diffusion model for single or sparse-view 3d object reconstruction." European Conference on Computer Vision. Cham: Springer Nature Switzerland, 2024.
\
[2] Deng, Zijun, et al. "MV-Diffusion: Motion-aware video diffusion model." Proceedings of the 31st ACM International Conference on Multimedia. 2023.
\
[3] Charatan, D., S. Li, and A. Tagliasacchi. "Sitzmann, V. pixelSplat: 3D Gaussian splats from image pairs for scalable generalizable 3D reconstruction." arXiv preprint (2023).
\
[4] Tang, Shengji, et al. "Hisplat: Hierarchical 3d gaussian splatting for generalizable sparse-view reconstruction." arXiv preprint arXiv:2410.06245 (2024).

---

### Meta-Review · Area_Chair_3hP3 · 2025-12-24

**Summary:**

This paper was flagged for ethics review due to plagiarism concerns from Reviewer ufbt, which where confirmed by AC Vkp4. I agree with the concerns.

Independent of the outcome of such a review, I recommend to reject the paper due to the general concerns the reviewers raised, which are given below. The scores were unanimously negative before the discussion phase and the authors did not provide answers to address the concerns.

**Reviewer Concerns:**

- Plagiarism concerns: Replicated content from VCR-GauS and MV-DUSt3R
- Generally outdated baselines and related work
- Missing clarity
- Lacking experimental rigor
- Lacking novelty

**Reviewer Scores:**

No changes since no author answer

---

### Decision · Program_Chairs · 2026-01-26

Reject